# Carotenoid metabolism strengthens the link between feather coloration and individual quality

Ryan J. Weaver [1], Eduardo S.A. Santos[2], Anna M. Tucker[3], Alan E. Wilson[4] & Geoffrey E. Hill[1]

Thirty years of research has made carotenoid coloration a textbook example of an honest signal of individual quality, but tests of this idea are surprisingly inconsistent. Here, to investigate sources of this heterogeneity, we perform meta-analyses of published studies on the relationship between carotenoid-based feather coloration and measures of individual quality. To create color displays, animals use either carotenoids unchanged from dietary components or carotenoids that they biochemically convert before deposition. We hypothesize that converted carotenoids better reflect individual quality because of the physiological links between cellular function and carotenoid metabolism. We show that feather coloration is an honest signal of some, but not all, measures of quality. Where these relationships exist, we show that converted, but not dietary, carotenoid coloration drives the relationship. Our results have broad implications for understanding the evolutionary role of carotenoid coloration and the physiological mechanisms that maintain signal honesty of animal ornamental traits.

[1] Department of Biological Sciences, Auburn University, 331 Funchess Hall, Auburn, AL 36849, USA. [2] BECO do Departamento de Zoologia, Universidade de São Paulo, São Paulo, SP 05508-090, Brazil. [3] School of Forestry and Wildlife Sciences, Auburn University, 3301 SFWS Building, Auburn, AL 36849, USA. [4] School of Fisheries, Aquaculture, and Aquatic Sciences, Auburn University, 203 Swingle Hall, Auburn, AL 36849, USA. Correspondence and requests for materials should be addressed to R.J.W. (email: rjw0019@auburn.edu)

ed, yellow, and orange carotenoid-based color displays are among the most widespread and conspicuous ornamental traits in animals. Carotenoid coloration is frequently an important criterion in mate choice[1,2], and researchers have found associations between carotenoid coloration and various measures of individual quality in studies of fish, reptiles, and birds[3–6]. On the basis of these observations, researchers hypothesized that assessment of the carotenoid coloration of prospective mates provides key information about individual quality and enables choices that increase fitness[7,8]. However, the mechanism that links carotenoid coloration to individual quality, and hence the specific information content of carotenoid displays, remains unresolved and contentious[5,9].

In addition to being colorful, carotenoids are thought to be relevant and essential molecules used in cellular processes such as immunocompetence and vitamin A synthesis, as well as anti-oxidants[10,11]. Understanding the connections between individual quality and carotenoid ornamentation demands an under-standing of the biochemistry of the pigments involved (Supple-mentary Note 1,[12,13]). Carotenoids cannot be synthesized by animals de novo; whether they are used as external colorants or serve functions in cellular pathways, all carotenoids must ulti-mately be obtained from the diet[14]. Moreover, animals typically ingest only the yellow carotenoid pigments (e.g., lutein and zeaxanthin; Fig. 1). To display carotenoid-based red coloration, most animals have to bio-convert yellow pigments to red[15,16]. The quality of an individual color display is a product of the biochemical pathways by which carotenoids are absorbed, transported, metabolized, and deposited[17], so it follows that these biochemical pathways are central in creating the connections between ornamental coloration and individual condition[7].

The dual role of carotenoids, both as colorants and potentially as important components of core cellular processes, is the foun-dation of the resource allocation trade-off hypothesis[9], which proposes that carotenoid coloration links to individual quality through tradeoffs in use of carotenoids for body maintenance versus ornamentation[18–20]. Under this hypothesis, carotenoids are needed both for body maintenance and for ornamentation, and thus only individuals with large stores of carotenoids or with low demands for body maintenance due to superior health can afford to allocate sufficient carotenoids for production of full ornamentation. The resource allocation trade-off hypothesis does not predict a difference in the signal content of coloration derived from metabolically modified versus dietary pigments.

Alternatively, carotenoid coloration may serve as a reliable signal of individual quality because the mechanisms involved in the metabolic conversion of carotenoid pigments (Fig. 1; Sup-plementary Note 1) are intimately linked to vital cellular path-ways[9,12,21]. This shared pathway hypothesis predicts that, regardless of the carotenoid resources that are available, disrup-tion of core cellular processes, and particularly cellular respira-tion, will reduce production of carotenoid ornamentation[7,22]. Moreover, under the shared pathway hypothesis, metabolism of dietary carotenoids for ornamentation should create stronger connections between cellular processes that give rise to quality and carotenoid-based signals because the pathways required for carotenoid metabolism are sensitive to the cellular environ-ment[12,23]. The process of transforming dietary carotenoids to ketocarotenoids (Supplementary Note 1) could be a key mechanism responsible for maintaining honesty from converted yellow and red carotenoid-based feather coloration, but this idea has not been rigorously tested.

Studies of the function and evolution of carotenoid colora-tion have frequently focused on birds and particularly on plu-mage. Sufficient studies have now been published to evaluate patterns among studies for associations between individual quality and feather coloration resulting from carotenoid pig-ments that emerge from fundamentally different biochemical

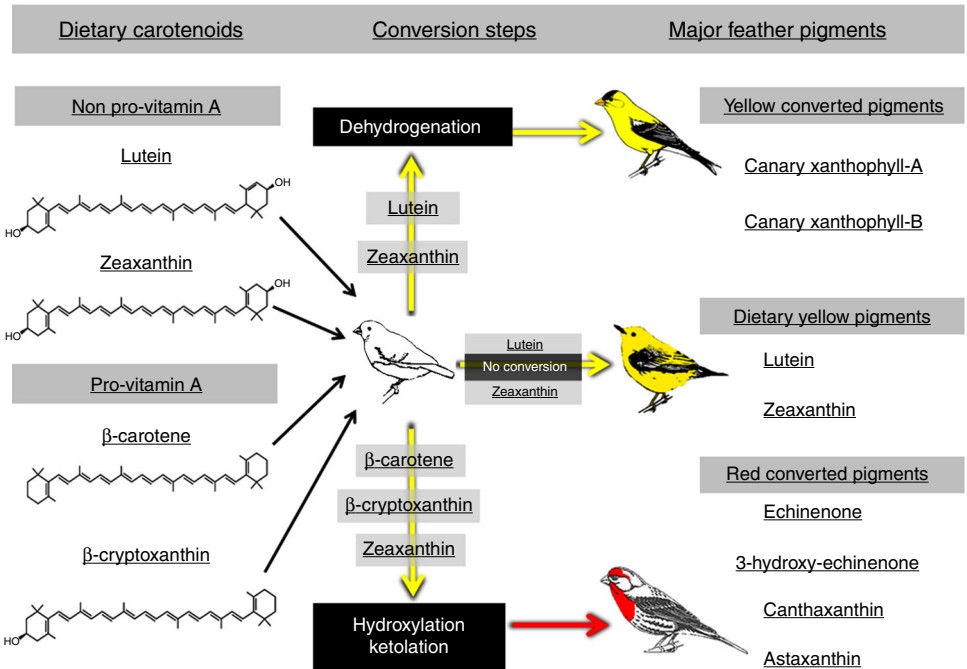

**Fig. 1** Carotenoids used for feather coloration. To produce feather coloration, passerine birds either use dietary carotenoids unaltered or use carotenoid pigments that are metabolically derived from dietary pigments. Depicted are the proposed metabolic pathways by which the common dietary carotenoids in the diets of passerine birds can be converted into the red and yellow ketolated carotenoids found in feathers[16,79,80]. Birds shown are representative of coloration from converted and dietary carotenoids. House finch (lower) use converted red carotenoids, American goldfinch (upper) display converted yellow carotenoids, and wood warblers (middle) display dietary yellow carotenoids. A few bird species directly ingest red carotenoids, such as astaxanthin, from their diet (not shown)

**Table 1 Results from mixed-effects multi-level phylogenetic meta-analyses that assume effect sizes within studies are correlated ($r = 0.5$)**

| Analysis | k | m | Mean (Zr) | Lower CI (2.5%) | Upper CI (97.5%) | $I^2$(%) | Heterogeneity Study (%) | Species (%) | Phylogeny (%) |
|---|---|---|---|---|---|---|---|---|---|
| Overall | 191 | 19 | 0.1609 | −0.0797 | 0.368 | 98.10 | 0.68 | 1.17 | 2.24 |
| Carotenoid type: | | | | | | 72.43 | | | |
| Converted | 92 | 12 | **0.263** | **0.0622** | **0.492** | | | | |
| Dietary | 99 | 7 | 0.089 | −0.1235 | 0.292 | | | | |
| Category (combined) | | | | | | 94.39 | | | |
| Condition | 35 | 12 | 0.0645 | −0.2184 | 0.352 | | | | |
| Immune and Oxidative | 42 | 10 | 0.1015 | −0.2107 | 0.362 | | | | |
| Parasite Resistance | 49 | 10 | 0.2433 | −0.074 | 0.510 | | | | |
| Reproductive and parental quality | 65 | 9 | 0.223 | −0.099 | 0.462 | | | | |
| *Category (Converted)* | | | | | | 72.4 | | | |
| Condition | 18 | 8 | 0.084 | −0.158 | 0.358 | | | | |
| Immune and Oxidative | 21 | 6 | 0.097 | −0.185 | 0.367 | | | | |
| Parasite Resistance | 30 | 8 | **0.435** | **0.174** | **0.688** | | | | |
| Reproductive and Parental Quality | 23 | 3 | **0.337** | **0.021** | **0.618** | | | | |
| *Category (Dietary)* | | | | | | | | | |
| Condition | 17 | 4 | 0.107 | −0.186 | 0.387 | | | | |
| Immune Function | 21 | 4 | 0.113 | −0.135 | 0.371 | | | | |
| Parasite Resistance | 19 | 3 | 0.009 | −0.241 | 0.283 | | | | |
| Reproductive and Parental Quality | 42 | 6 | 0.095 | −0.183 | 0.311 | | | | |

Category (combined) represents estimates from a meta-analytic model with life history trait category as a predictor, while Category (Converted) and Category (Dietary) represent estimates from a meta-analytic model with an interaction between life history trait category and type of carotenoid as predictor variables. Effect sizes in bold are considered to be statistically significantly different from 0, as the 95% credible interval did not overlap 0
k number of effect sizes, m number of species

processing. In this quantitative synthesis, we categorize carotenoids used for feather coloration as 'dietary' if the pigments are present in food and deposited in feathers without further modifications (Fig. 1). Pigments in this category are absorbed, transported, and deposited but they undergo no metabolic conversion. The two most common dietary pigments used as colorants in birds are lutein and zeaxanthin. Alternatively, we categorize carotenoids as 'converted' if they are derived from dietary carotenoids that are metabolically oxidized internally by the bird to form ketocarotenoids before deposition to feathers (Fig. 1; Supplementary Note 1). Common examples of converted carotenoids include echinenone, canthaxanthin, and canary xanthophylls[16].

Previous reviews of condition dependence of carotenoid pigmentation in birds have not considered whether the biochemical processes involved in carotenoid pigmentation—and specifically whether or not dietary carotenoids are metabolically converted—affects the relationship between coloration and quality[6,24,25]. We hypothesized that the strength of the relationship between coloration signal and individual quality is dependent on whether the color display involved metabolic conversion of dietary carotenoids. We predicted that if the mechanisms of carotenoid metabolism are linked to basic cellular function that comprise individual quality[9,26], feather coloration requiring carotenoid metabolism would have a stronger positive relationship with measures of individual quality than feather coloration derived from deposition of unaltered dietary carotenoids[12,23]. Additionally, metabolism of carotenoids requires the maintenance of enzyme systems (Supplementary Note 1;[27]) and perhaps additional transport of carotenoids[13] that may create stronger links between coloration and system performance for ornaments produced from converted pigments versus unaltered dietary pigments.

We tested our hypothesis that carotenoid metabolism strengthens the link between feather coloration and individual quality using meta-analysis. We quantitatively synthesized published results on the relationships between individual quality and plumage coloration of passerines produced via dietary versus converted carotenoids. Overall, we find that, without partitioning between carotenoid types, carotenoid-based feather coloration is positively associated with individual quality. However, including carotenoid type as a predictor reveals that converted, and not dietary, carotenoids are driving the relationship between coloration and quality. Thus, the physiological processes involved in carotenoid metabolism may be an important mediator in maintaining honesty from carotenoid-based coloration.

## Results

**Final data set of included studies and assessment of publication bias.** The final data set included 191 effect size estimates (Supplementary Data 1) from 50 published studies of 19 passerine bird species (Supplementary Data 2). Detailed results from models assuming effect sizes are correlated are listed in Table 1. Results from models assuming no effect size correlations are reported in Supplementary Table 1. Mean effect size estimates and 95% credible intervals (CI) are reported unless otherwise noted. Mean estimates with credible intervals that do not include zero are a statistically significant effect at $\alpha = 0.05$. Using sampling variance as a predictor, results from the Egger's regression did not imply funnel plot asymmetry from the residuals of the overall meta-analytic model (Supplementary Fig. 1; weighted linear regression: $t_{189} = 1.98$, $p = 0.05$).

**Carotenoid coloration as an honest signal of quality.** Consistent with previous quantitative and qualitative reviews that did not account for carotenoid type, we found a small positive correlation

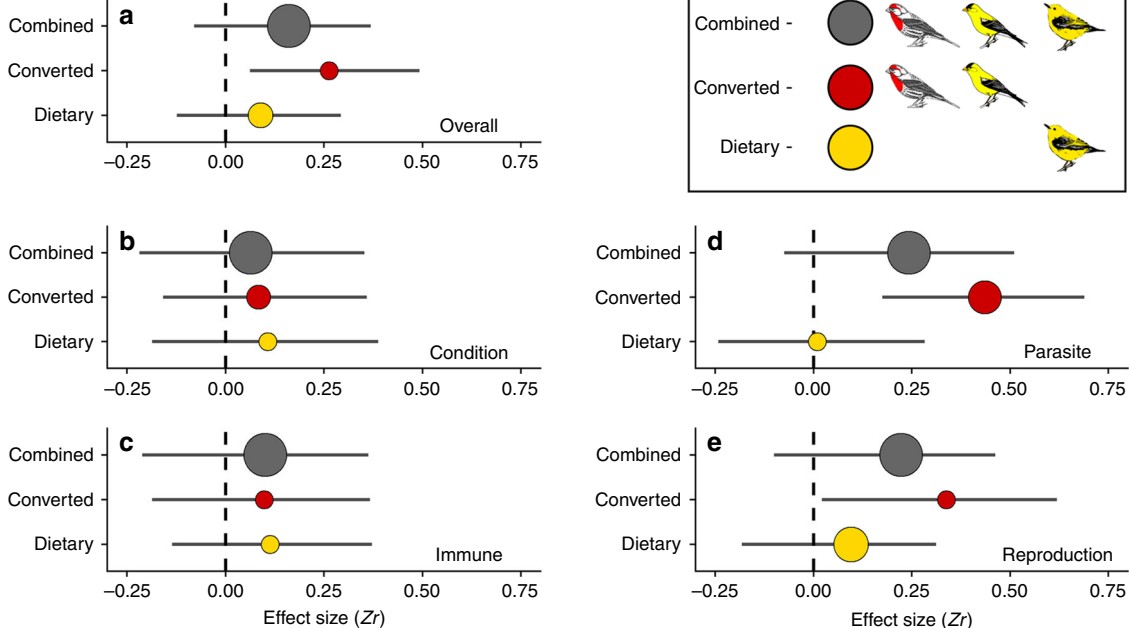

**Fig. 2** Results of the multilevel phylogenetic meta-analyses. The weighted mean correlation (Zr) between feather color richness and measures of individual quality. **a** The strength of the association was calculated for all published studies without consideration of the carotenoid type in the feathers of the study bird (Combined, gray circles, number of effect sizes ($n$) = 191), for only studies of bird species with plumage coloration derived from converted carotenoids (red circles, $n$ = 92), and for only studies of bird species with plumage coloration derived from dietary carotenoids (yellow circles, $n$ = 99). Proxies of quality were divided into (**b**), body condition (overall: $n$ = 35, converted: $n$ = 18, dietary: $n$ = 17), (**c**) immune and oxidative physiology (overall: $n$ = 42, converted: $n$ = 21, dietary: $n$ = 21), (**d**) parasite resistance (overall: $n$ = 49, converted: $n$ = 30, dietary: $n$ = 19), and (**e**) aspects of parental and reproductive quality (overall: $n$ = 65, converted: $n$ = 23, dietary: $n$ = 42). Circle size is inversely proportional to the variance of the mean effect size. Horizontal lines are error bars that represent 95% credible intervals. Effect size estimates with credible intervals that do not include zero are statistically significant effects ($\alpha$ = 0.05)

between richness of coloration and individual quality overall, but note that the 95% CI slightly overlaps zero (weighted linear regression: Zr = 0.161, 95% CI: −0.079 to 0.368; $I^2$ = 98.10%). However, when we parsed the data between species with plumage coloration resulting from converted versus dietary carotenoids, we found that the relationship between richness of coloration and individual quality held only for species that used converted pigments (weighted linear regression: Converted: Zr = 0.263, 95% CI: 0.062–0.491; Dietary: $Zr$ = 0.089, 95% CI: −0.123 to 0.292; Fig. 2a). On average, we found that dietary carotenoid coloration is a weaker predictor of overall quality than converted carotenoid coloration (weighted linear regression: $\beta_{(Dietary)}$ = −0.158, 95% CI: −0.388 to 0.097), though the credible intervals from this comparison slightly overlap zero.

**Body condition**. Measures of body condition were not reliably positively correlated with carotenoid-based feather coloration (weighted linear regression: Zr = 0.065, 95% CI: −0.218 to 0.352, Fig. 2b), and we found no clear difference in the relationship between carotenoid types (weighted linear regression: Converted: Zr = 0.084, 95% CI: −0.158 to 0.358; Dietary: Zr = 0.107, 95% CI: −0.186 to 0.388).

**Immune function and oxidative physiology**. On average across all measures of immune function and oxidative physiology, we found that feather coloration was not a reliable predictor (weighted linear regression: Zr = 0.102, 95% CI: −0.210 to 0.362; Fig. 2c) with no clear difference in the relationship between carotenoid types (weighted linear regression: Converted: Zr = 0.098, 95% CI: −0.186 to 0.367; Dietary: Zr = 0.113, 95% CI: −0.135 to 0.371). When we treated these two categories separately, the results were qualitatively similar (Supplementary

Table 2); neither dietary nor converted carotenoids were robust signals of immune function or measures of oxidative physiology.

**Parasite resistance**. When no distinction was made between carotenoid types, richness of coloration was positively correlated with parasite resistance (weighted linear regression: Zr = 0.243, 95% CI: −0.074 to 0.510). When we added carotenoid type as a moderator to our model we found that coloration from dietary carotenoids was weakly correlated with parasite resistance (weighted linear regression: Zr = 0.009, 95% CI: −0.242 to 0.371; Fig. 2d). In contrast, coloration from converted carotenoid-based plumage was strongly positively associated with parasite resistance (weighted linear regression: Zr = 0.435, 95% CI: 0.175–0.689; Fig. 2d) and was a more reliable signal than coloration from dietary carotenoid-pigmented feathers (weighted linear regression: $\beta_{(Dietary)}$ = −0.420, 95% CI: −0.692 to −0.162). When measures of parasite resistance were pooled with the immune category, our model return qualitatively similar results (Supplementary Table 3); effect sizes from dietary coloration were not robustly associated with measures of immune function, but effect sizes from converted coloration strongly correlated with measures of immune function.

**Reproductive and parental quality**. We found that, overall, carotenoid coloration positively correlated with aspects of reproductive and parental quality (weighted linear regression: Zr = 0.223, 95% CI: −0.099 to 0.462; Fig. 2e), but that the pattern was driven by converted carotenoid feather coloration (weighted linear regression: Converted: Zr = 0.337, 95% CI: 0.021–0.618; Dietary: Zr = 0.0.09, 95% CI: −0.183 to 0.312). However, converted feather color was only a marginally better predictor of

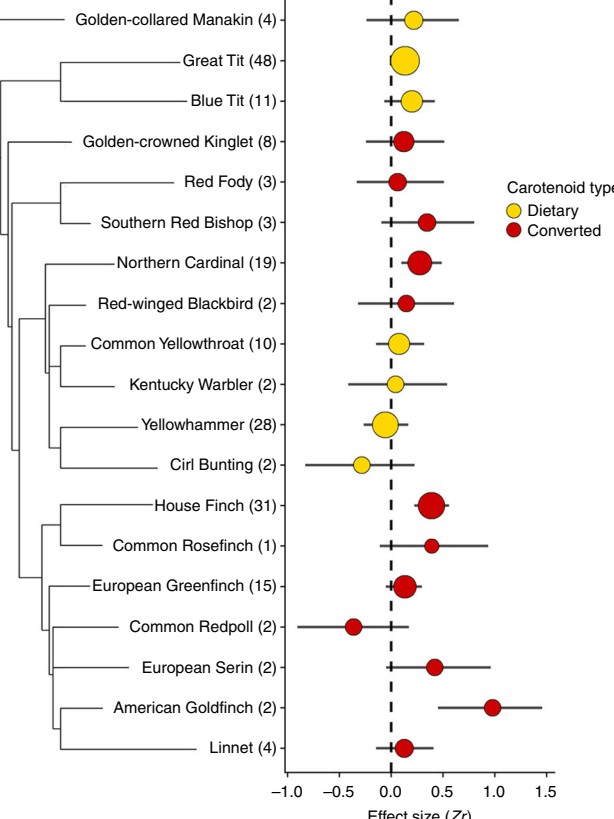

**Fig. 3** The contribution of each species to the overall relationship between feather color richness and measures of individual quality. Number of effect sizes per species are in parentheses. Circles represent weighted mean correlation (Zr) effect sizes for species with plumage coloration from converted (red) and dietary carotenoids (yellow). Circle size is inversely proportional to the variance of the mean effect size. Horizontal lines represent 95% credible intervals. Effect size estimates with credible intervals that do not include zero are statistically significant effects ($\alpha = 0.05$)

quality than dietary feather color (weighted linear regression: $\hat{\beta}_{(Dietary)} = -0.227$, 95% CI: −0.536 to 0.077).

**Species-level analysis.** The strength of the relationship between feather coloration and combined measures of individual quality was dependent on the individual species (Fig. 3). However, we found only a weak phylogenetic signal among carotenoid types ($I^2 = 2.24\%$) in the overall analysis (Table 1).

**The role of sex on effect size estimates.** We found that the sex from which effect sizes were calculated did not influence the magnitude of the meta-analytic mean (weighted linear regression: Dietary female vs male: $\beta = 0.008$, 95% CI: −0.140 to 0.170; Converted female vs male: $\beta = 0.101$, 95% CI: −0.07 to 0.29).

**Residual heterogeneity.** Heterogeneity in our overall meta-analytic model was high ($I^2 = 98.12\%$; see ref. [28] for benchmarks). Including carotenoid type as a moderator in our overall model reduced heterogeneity ($I^2 = 72.43$; Table 1). More variance was partitioned from the model by the addition of quality category and carotenoid type as model moderators ($I^2 = 72.40$; Table 1). A high degree of heterogeneity remained after we included carotenoid type and quality category as moderators in the model indicating that other variables not included in this analysis may

be influencing the relationship between feather coloration and aspects of individual quality[29].

## Discussion

Carotenoid coloration is among the most frequently cited examples of a condition-dependent signal of quality assessed during mate choice[1,30], but the relationship between various measures of quality and different forms of carotenoid coloration is not consistent across studies[31-33]. We hypothesized that some of the noise in the comparative data might be due to a failure to consider whether the color display under study was derived from unmodified dietary pigments or pigments that had been metabolically converted from dietary pigments. We predicted that feather displays derived from converted carotenoids would be better signals of quality if carotenoid metabolism creates stronger connections between color displays and the cellular processes that underlie the state of condition in the animal.[12,21] The patterns we observed in our meta-analyses indicate that converted carotenoid feather coloration indeed had stronger relationships with measures of individual quality than dietary carotenoid feather coloration.

The results of our meta-analyses show that whether the carotenoid coloration is dietary or converted is an important predictor of the likelihood that a plumage color display will be correlated to a measure of individual quality. When we assessed all studies of carotenoid feather coloration, without regard for whether the pigments were dietary or converted, there was a non-significant trend across studies for a positive association between carotenoid coloration and individual quality (Fig. 2a). When we examined whether the type of carotenoid (converted or dietary) used for plumage coloration had an effect on the strength of the relationship between carotenoid coloration and condition, we found that there was a strong and statistically significant correlation between measures of quality and coloration for birds that deposit converted carotenoids to feathers. In contrast, there was much weaker condition dependency for birds that deposited only dietary carotenoids to feathers, and the relationship was not significantly different from zero. We did not observe a clear phylogenetic pattern in the strength of the relationship between coloration and quality (Fig. 3). These results support our hypothesis that carotenoid metabolism is a meaningful factor that predicts the strength of correlation between coloration and measures of individual quality.

Our findings have important implications for understanding the mechanisms that link carotenoid coloration and individual condition. Endler[34] first proposed that carotenoid coloration in animals could serve as an honest signal of quality because carotenoids are scarce resources that are challenging to accrue. Experiments with guppies (*Poecilia reticulata*)[35] and house finches (*Haemorhous mexicanus*)[36] showed that depriving individuals of carotenoids during ornament production caused loss of color, but whether resource limitation was the basis for signal honesty in natural environments remained contentious[37-39]. The patterns revealed in our meta-analysis suggest that honest signaling arises from more than simply access to carotenoid resources, whereby all forms of carotenoid pigmentation should serve as signals of quality. Rather, the strong association between display of converted carotenoid pigments and individual quality suggests that something about the process of carotenoid modification creates a link between pigmentation and individual condition[7,9,21,22].

Perhaps one of the most ubiquitous statements made of carotenoids is their capacity to serve as powerful antioxidants and immune boosters[10,40,41]. However, the specific roles that carotenoids might play in vivo in free radical scavenging or immune

system function are uncertain and contentious[42–45]. A previous meta-analysis by Simons et al.[6] examined the relationship between carotenoid content and multiple measures of immunocompetence and oxidative stress in birds to test the idea that carotenoid content or coloration is an honest signal of these qualities. They found that circulating carotenoid levels in the plasma related to only two out of seven measures: PHA response —a commonly used skin swelling assay of immunocompetence— and antioxidant capacity. Additionally, carotenoid coloration of feathers or skin was not a reliable signal for six out of seven proxies of immune function or oxidative stress[6]. Our results are generally consistent with those of Simons et al.[6] in that feather coloration was weakly correlated with measures of immune function and oxidative physiology, and was not a robust signal on average (Fig. 2c). Furthermore, whether coloration was derived from dietary or converted carotenoids had no effect on the honesty of coloration as a signal of immune function. The complex pathways involved in innate and acquired immunity may obscure meaningful interpretation of immunocompetence when it is measured only as the relative abundances of different types of white blood cells in circulation.

Unlike measures of immune function, studies on internal or surface parasites in birds benefit from a simple methodology that is straightforward to interpret. One of the earliest and widely cited examples of honest signaling of parasite resistance from feather coloration was Hamilton and Zuk's[46] study that proposed that 'bright' feather coloration evolved to signal heritable parasite resistance. This seminal study served as the cornerstone for new ideas about why ornaments are assessed during mate choice. We found that birds with converted carotenoid feather coloration had a strong relationship with parasite resistance. In contrast, birds that directly deposited carotenoids from their diet showed a weak and unreliable correlation with parasite resistance (Fig. 2d). The mechanisms involved in the transformation of dietary carotenoids might act as a signal of the underlying genetic or physiological mechanisms that provide parasite resistance[47].

The stronger associations between reproduction and plumage color derived from converted versus dietary pigments could be a consequence of stronger associations between individual condition and ornamentation produced by converted versus dietary carotenoids. The proxies for reproductive and parental quality that were used in this study are a composite of the strategies and performances of both the ornamented male bird and its mate[48–50]. Mate choice for feather coloration should be stronger when the signal of quality is stronger, so the stronger association between reproduction and converted carotenoids could arise through stronger choice for such signals. Moreover, if the stronger preference for coloration resulting from converted carotenoids means that such birds attract higher quality mates then color based on converted carotenoids would be more strongly linked to reproductive success. For example, in many species of perching birds, males directly participate in nesting so there should also be direct effects of male condition on reproductive success. Any of these factors could have contributed to the stronger associations of color produced by converted carotenoids and reproduction.

The proxies for quality used in nearly every study of carotenoid coloration are rooted in the ability to efficiently produce ATP via oxidative phosphorylation at a level necessary to meet physiological demand. Thus far, the mechanism by which carotenoid coloration links to quality has remained speculative. Direct measures of both the allocation of specific pools of carotenoids to physiological function versus ornament production and the effects of direct manipulation of mitochondrial performance on carotenoid production are needed to definitively test the resource allocation and shared pathway hypotheses.

The patterns that we revealed though our meta-analysis indicate that feather coloration from converted carotenoids are better indicators of individual condition than dietary carotenoids. One could reasonably extrapolate from this observation to propose that metabolism of feather carotenoids evolved in response to mate preferences for the best indicators of condition. But the conversion of yellow dietary to yellow converted pigments, which present the same visual display, presents an enigma. Why don't males that display with converted yellow pigments just skip the conversion and load up on dietary pigments? Badyaev et al.[51] examined patterns of feather growth and coloration and hypothesized that metabolized carotenoids caused less stochastic variation in feather structure than dietary carotenoids. In other words, dietary pigments can affect the integrity of feathers and perhaps the stochastic variation they engender can undermine the quality of the sexual signal. In light of our finding that converted pigments provide a better indicator of condition, it will be interesting to further test such hypotheses for how developmental pathways might favor converted over dietary carotenoids.

Using a meta-analysis and meta-regressions of bird studies that investigated the association between carotenoid-based feather coloration and individual condition, we found support for the hypothesis that this type of plumage coloration serves as an honest indicator of individual quality[52]. The strength of this signal, however, is greater for species that convert dietary carotenoids to ketolated carotenoids before utilizing them for feather coloration.

Our results are not consistent with the resource allocation hypothesis because dietary carotenoids are the pigments primarily implicated in immune enhancement and free radical scavenging, and yet they were not consistently associated with proxies for individual quality. Our results are consistent, however, with the hypothesis that conversion of dietary carotenoids acts to strengthen the relationship between feather coloration and measures of quality and may be an important mechanism in maintaining honesty from carotenoid-based ornaments.

Feathers are not the only structures that are colored by carotenoids. Many bird, reptile, and fish taxa exhibit bare-part carotenoid color signals[53–56]. Future work could focus on understanding the differences in signal honesty from bare-part carotenoid coloration among different taxa to test alternative predictions of our hypothesis that carotenoid metabolism strengthens the link between coloration and individual quality.

## Methods

**Literature search**. A literature search of the *Web of Science* and *Google Scholar* databases using alternative spellings and combinations of the keywords, including "carotenoid", "color", "condition", "signal", "feather", and "quality", returned over 1600 potentially eligible published articles. This search was based on the Preferred Reporting Items for Systematic reviews and Meta-Analyses statement (PRISMA[57]; Supplementary Fig. 2). We also considered as potentially eligible articles, data from studies included in published meta-analyses of bird coloration[6,24] and references therein. The online database search was last performed on 11 April 2017 on titles, abstracts and keywords in both databases. We did not request the unpublished data sets from colleagues, because of the risk of biasing the estimates of effect size (see ref. [58]).

It is important to note that carotenoid coloration is not limited to feathers; indeed, carotenoid coloration from bare-parts, including beaks, legs, and combs of birds has been the focus of excellent research on the evolution of honest signaling[53,59–61]. However, to best address how carotenoid metabolism affects honest signaling, we focused our study on a single trait, feather coloration, to avoid confounding biological factors such as blood flow[62], carotenoid esterification[61], and differences in requisite enzymes[27,63] that are relevant to bare-part coloration, but not feather coloration. These factors could obscure meaningful interpretation of the relationship among coloration measures, carotenoid metabolism, and individual quality from bare-part coloration.

Because we were interested in the signal content of carotenoid-based plumage, we focused on studies that quantified feather color using standardized color metrics (see 'Color metrics' below) of natural (i.e., un-supplemented) adult bird color levels. Therefore, we excluded studies from our meta-analysis for any of the

**Table 2 Commonly used proxies for quality to evaluate the relationship between carotenoid-based feather coloration and individual quality**

| Category | Metric | Relationship with quality |
|---|---|---|
| Condition | Body mass | + |
|  | Size-adjusted body mass | + |
|  | Ptilochronology | + |
| Immune and oxidative physiology | Anti-oxidant capacity | + |
|  | Antibody count | + |
|  | Disease survival | + |
|  | Environmental pollutant | − |
|  | HL ratio | − |
|  | Pathogen challenge | − |
|  | Oxidative damage | − |
|  | PHA response | + |
|  | WBC count | + |
| Parasite | Load | − |
|  | Resistance | + |
| Parental and reproductive quality | Clutch size | + |
|  | Feeding rate | + |
|  | Fledgling number | + |
|  | Fledgling success | + |
|  | Lay date | − |
|  | Nest attentiveness | + |

+ indicates that this measure has a positive correlation with individual quality.− indicates that this measure has a negative correlation with individual quality

following reasons: only coloration of non-feathered structures was measured (e.g., wattles, legs, beaks); a non-passerine species was studied; only plasma concentrations of carotenoids were measured; or only nestling or juvenile coloration was studied. We did not include measures of feather brightness as it is sensitive to factors unrelated to pigmentation (see below). Additionally, to be included in our meta-analysis, studies must have investigated at least one of the following proxies of individual quality: (1) nutritional condition, (2) immune function or oxidative capacity, (3) parasite resistance, or (4) reproductive or parental quality (Table 2).

**Data extraction and coding**. We used the correlation coefficient, Pearson's $r$, as the effect size metric to describe the association between measurements of feather coloration and aspects of individual quality. The effect size metric was extracted directly from each study when available ($n = 9$). For cases in which studies did not provide Pearson's $r$ ($n = 41$), the reported test statistics ($F$, $\rho$, $\chi^2$, $\tau$, $t$, and means and standard deviations or standard errors) were used to estimate $r$[64]. Pearson's $r$ was transformed to Fisher's $Z$ for statistical analyses to meet normality assumptions of linear models. Researchers commonly refer to $r = 0.1$, 0.3 and 0.5 as small, medium, and large effect sizes ([65]; these benchmarks are equivalent for Zr values), respectively. The sign of the correlation coefficient was changed in some cases to facilitate comparisons across metrics of individual quality; for example, some measures— such as parasite load—were often negatively correlated with richness of coloration but indicated a positive relationship with quality[66].

Original studies often include multiple effect size estimates because of the measurement of several life history traits associated with individual condition or performance (*e.g.*, immune function or fledging success) and/or from different color metrics (see below). When more than one effect size is reported these data points cannot be assumed to be independent. To deal with this issue, we ran multi-level, random-effects meta-analytic models that allowed us to include multiple, non-independent effect sizes per study. This was accomplished by including the study identity and species identity as random effects[67].

**Carotenoid type**. Coloration of feathers was categorized as dietary if feather carotenoids identified for that species were only those that are typically found in passerine diets (e.g., lutein or zeaxanthin)[16,68]. Carotenoids were categorized as converted if feather carotenoids identified for that species were those that are oxidative products of dietary carotenoids, typically 4-keto-carotenoids and canary xanthophylls (Supplementary Note 1 [16,68]). We referenced published studies that characterized the carotenoids present in feathers of each species included in our meta-analysis (Supplementary Table 4). For instances in which we could not identify the feather carotenoids of a particular species ($n = 3$), we used the

published data from a sister-species of the same genus that displayed the same feather color.

**Color metrics**. Standing variation in richness of color between individuals is requisite for assessing quality from carotenoid-based ornaments. Common metrics used to quantify this variation in feather reflectance include comparisons to standard color charts (e.g., Munsel), calculations of hue, chroma and brightness or composite metrics such as principal components (PCA) from spectrophotometer data or digital photographs. Hue describes the unique spectral color (e.g., "red", "orange", "yellow") and chroma describes the saturation or spectral purity of the color display relative to total reflectance across the visible range of the electromagnetic spectrum[69]. We extracted the response of all color variables used to assess the relationship between color and a measure of quality (e.g., condition, immune function) from each study. We did not include measures of brightness in our analyses because it is strongly influenced by the physical structure of the feather which may be altered by abrasion and wear and is difficult to interpret for carotenoid content[70].

**Meta-analysis technique**. To evaluate the strength of correlation between all measures of individual quality and overall feather color richness, we first used a Bayesian mixed-effects meta-analytic model without distinguishing between carotenoid types (i.e., "Combined"; intercept only model). To further partition heterogeneity, we conducted a Bayesian meta-regression analysis with carotenoid type included as a categorical moderator to test the effect of the source of ornamental coloration (either dietary or converted carotenoids) on the overall relationship between color and quality. We then performed a second meta-regression analysis to test the strength of association of each quality category with both dietary and converted carotenoid-based plumage. We used a third meta-regression model to quantify the relationship between coloration and quality for each bird species examined in the data set. Lastly, we used a fourth meta-regression model that included an interaction between the carotenoid-type and sex in the relationship between color measures and quality. We used MCMCglmm package[71,72] in program R version 3.3.0[73] to conduct the Bayesian mixed-effects meta-analyses (Supplementary Data 3).

To adequately correct for non-independence, it is necessary to model the correlations among effect sizes. For instance, effect sizes for immune response and fledging success are likely to be correlated when the traits were measured from the same group of individuals. Unfortunately, such correlations are almost never reported in the original studies, and are seldom accounted for in meta-analytic models (but see ref.[74]), potentially biasing general findings. Thus, to conservatively account for such correlations, we report multi-level models that assume all correlations to be 0.5. Additionally, in Supplementary Table 1, we report the results of the same models assuming correlations of zero. We note that qualitatively the results are very similar among these meta-analytic models.

We also accounted for the evolutionary history of the bird species used in the study by including a phylogenetic random effect to the models. Our meta-analytic models used avian phylogenetic trees with the Ericson backbone from ref. [75]. To account for uncertainty in the phylogenetic reconstruction, we used a sample of 1000 trees, so that each tree was sampled at iteration, $t$. We calculated phylogenetic heritability, $H^2$, as an index of phylogenetic signal. $H^2$ can be defined as the proportion of phylogenetic variance in relation to the sum of all other variance components, with the exception of sample error variance. When the unit of analysis is at the species level, $H^2$ is equivalent to Pagel's $\lambda$[76].

For all multi-level models, we used parameter expanded priors ($V = 1$, $nu = 1$, alpha.mu $= 0$, alpha.$V = 1000$) for all random effects. We used a combination of total iterations and burn-in period so that the posterior distributions consisted of 1000 samples for the model parameters. We report point estimates from the models based on the posterior means, and considered moderator estimates to be statistically significant if the 95% credible interval (95% CI) did not overlap zero. We calculated an modified version of the $I^2$ statistics (following ref. [67]) to estimate heterogeneity in multi-level meta-analytic models. This procedure allocates the proportion of variance not attributable to sampling variance to the random factors of the model. In our models, these are the variance in effect sizes due to the following four components: phylogenetic relatedness, differences among species, differences among studies, and differences residual variation (within-study variation). Adding the total variation (percentages) from each of these components yields the original $I^2$ proposed by ref. [28].

To test for potential publication bias, we visually evaluated funnel plot asymmetry of effect size as a function of sample size. Then, we conducted an Egger's regression[77] to test statistically for publication bias. The analyses of bias for the multi-level models was conducted on the meta-analytic residuals (see ref. [67]), as this safeguards that we meet the assumption of independence.

**Code availability**. The code used to perform the meta-analyses that support the findings of this study are available in figshare repository with the identifier: https://doi.org/10.6084/m9.figshare.5675431.v2 (ref. [78]).

**Data availability**. The effect size data that support the findings of this study are available in figshare repository with the identifier: https://doi.org/10.6084/m9.

figshare.5675431.v2 (ref. [78]). The authors declare that all the data supporting the findings of this study are available within the paper and its supplementary information files.

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

## Acknowledgements

Thanks to Shinichi Nakagawa for sharing R script files. Feedback from the Hill-Hood-Wada lab group improved previous drafts of this manuscript. G.E.H. was supported by the Department of Biological Sciences and the College of Science and Mathematics at Auburn University. A.E.W. was supported by the School of Fisheries, Aquaculture, and Aquatic Sciences and the College of Agriculture at Auburn University.

## Author contributions

R.J.W., G.E.H. and A.E.W.: Designed the study. R.J.W.: Collected the data. R.J.W., A.M.T. and E.S.A.S.S.: Performed the meta-analyses. R.J.W. and G.E.H.: Wrote the first draft of the manuscript, and all authors contributed substantially to revisions.

## Additional information

**Competing interests:** The authors declare no competing financial interests.

