## [Peer Review File · Nature Communications]

Reviewers' comments:

Reviewer #1 (Remarks to the Author):

Review of NCOMMS-17-21773, Carotenoid metabolism strengthens the link between feather coloration and quality: a meta-analysis

Overview: In this paper, the authors assess whether metrics related to quality are more closely associated with dietary, metabolically unconverted carotenoids, or metabolically converted carotenoids in the feathers of songbird species. By performing a meta-analysis, they find support for the hypothesis that there are more robust relationships between metabolically converted carotenoids and quality than unchanged, diet-derived carotenoids. This work is interesting, and the authors are absolutely correct that the often-discussed but still contentious issue of HOW carotenoids relate to quality is an important question. I do have some questions/concerns/suggestions that I outline below.

Line 70: Relatively irrelevant to the authors' arguments, but what about carotenoids from maternal effects (e.g., transfer via yolk)?

Line 72: This is one of my larger concerns – I'm unclear why bare parts were dismissed for no explicit reason. In fact, inclusion of bare parts couldn't be more timely; please see Erik N. K. Iverson and Jordan Karubian (2017) The role of bare parts in avian signaling. *The Auk*: July 2017, Vol. 134, No. 3, pp. 587-611. <https://doi.org/10.1642/AUK-16-136.1> Inclusion of carotenoid-based bare part coloration would 1) increase sample size and thus generalization of the broader question, and 2) greatly expand the phylogenetic breadth of samples. All but two of the studies in this meta-analysis were songbirds, and yet some of the seminal work on carotenoid-based coloration was with non-passerines. Either the scope of the work needs to be explicitly restated, or bare-parts should be included. As is, it's hard for me to accept a general question regarding this study (see lines 99-101: "Specifically, we hypothesized that the strength of the relationship between coloration signal and individual quality is dependent on whether or not the color display involved metabolic conversion of dietary carotenoids.") There's no reason this hypothesis shouldn't include bare parts.

Line 82: It was unclear whether or not the authors were restricting their study to songbirds, as opposed to all birds.

Figure 1. Please include information on whether marine birds may acquire astaxanthin from their diets, though.

Line 106: I found this line to be too vague. What is meant by system performance? Later, the authors also talk about cellular respiration (line 377), which is fine. But the language kept on switching. Are system performance, cellular pathways (line 318), and cellular processes (lines 288 and 322) all talking about the same thing? During these arguments, I kept on wondering if the authors were doing a good job about seeing the forest rather than the trees, or if they were engaging in hand-waving. I think tightening up the language may shift readers toward the former.

Line 114: What about British versions of spelling colour and colouration?

Line 121: I think justifying why focusing ONLY on plumage-based colors is critical. Or – what would be even better – expanding to include bare part-based coloration.

Line 124: I didn't follow this. Why was measuring plasma concentrations of carotenoids a reason for exclusion? Did authors of papers used in the meta-analysis need to identify carotenoids in feathers, because I didn't see that statement. What about carotenoid IDs from other tissues (e.g., liver)? Did authors have to ID carotenoids in that specific study for it to be included, or did just one lab group have to ID the carotenoids in a species for any study on that species to be included? More clarity is required here.

Table 1. Oxidative physiology cannot be used interchangeably with immune function. I fully disagree with the authors' inclusion of AO capacity and OD in the Immune category. I'd also note that their main citation on this front (Simons et al. 2012) also separates these two factors (immune function and oxidative stress).

Line 181: findings

Table 2: The PDF version doesn't look right (I can't see all of the categories). Please check.

Line 261: Amounts?

Lines 286-289: It's a statement like this that screams for the inclusion of bare part coloration.

Lines 316-326: Some more precision here might help sell this idea. Does this apply to all cells? All organs? At all time points in the animal's life? I think I get the general idea, but would this apply only to the cells doing the converting of carotenoids? And what would those cells have to do with parasite resistance (but not immune function)? Statements like lines 320-323 sound like an a priori prediction for a positive relationship between metrics of immune function and coloration. At the very least, this statement wouldn't disambiguate between immune function and parasite resistance.

Lines 341-347: Please be clear the oxidative physiology isn't a sub-category of immune function.

Lines 388-393: Based on which results? I'm not sure why metrics of immune function are not related to quality, but parasite resistance is.

Reviewer #2 (Remarks to the Author):

The authors present a meta analysis of studies investigating the association between the

richness of carotenoid colouration of bird plumage, and measures of individual quality. Overall, they find a moderate-sized, positive association that does not statistically differ from zero. However, when the data are partitioned by carotenoid type (whether dietary carotenoids are directly deposited in plumage or first metabolised and converted to ketocarotenoids) and by the measure of individual quality used, there is a strong positive association between converted (but not dietary) carotenoid colouration and measures of parasite resistance and parental/reproductive performance. The authors argue that the metabolic conversion of carotenoids could act to maintain the honesty of colour signals, which seems plausible though it would be great to see this argument developed in even greater detail if possible. Interestingly, the analysis finds no associations between carotenoid colouration and either body condition or immune function, contra popular hypotheses for the signalling function of carotenoids.

The study is a solid and competent analysis of an interesting body of literature, and the contrast it reveals between dietary and converted carotenoids seems like a promising avenue for further research. I feel that this could also be developed further in the manuscript. What was lacking for me in the discussion was consideration of why some species convert carotenoids before deposition and others do not: what selects for ornamentation with dietary carotenoids if these are not signals of quality; and why convert carotenoids if they can be directly deposited in the plumage?

More generally, I think the manuscript would benefit from a more thorough treatment of the theoretical background to the study question – at the moment the introduction seems a bit light and a lot of helpful concepts only appear in the discussion. More on the main hypotheses around carotenoids as signals would be especially valuable, given you want to highlight the importance of biochemical and physiological mechanism to these hypothesised selective processes.

The intro itself could be better organised overall. I would suggest starting with carotenoids (e.g. something like the current lines 65-70) before the current first paragraph, and moving the section that starts on line 86 up to merge with the current first paragraph. This might help to clarify your statements on the state of the literature – currently, the first para implies that there is general support for the hypothesis that carotenoid colouration signals individual quality, but the mechanisms for this are disputed, while line 92 suggests that the relationship between carotenoid expression and quality is not generally supported. You could then merge the section line 72-84 with line 94-107. Finally, the specific predictions on line 101-107 are not immediately intuitive to me and would benefit from additional explanation.

Although the intro and methods do not mention sex differences in colour at all, you start the discussion with reference to the idea that carotenoid colouration in males is a signal used by females to assess mate quality. Apart from this being rather late in the manuscript to introduce new concepts, I am curious whether the studies included in your dataset used species that display sex-specific colouration, or even focused on males. Did the majority of studies measure carotenoid colour and quality in males only? If not, then sex differences seem like a possible source of heterogeneity – could you account for this in your analysis?

Looking at Table 1, there are a couple of 'quality' proxies that seem quite specific to female fitness (e.g. in the Reproduction category, clutch size or lay date) which suggests to me that at least some studies measured females, and I wonder if you might find differences in the type of carotenoid (dietary or converted) in species where males only, or both sexes, express the colour ornamentation.

Altogether, I think this is an interesting study but it would benefit from some thoughtful editing to better set up the background to the study, and provide more solid logical and indirect support for your interpretation of the influence of carotenoid type.

Minor:

L58 Referring to 'the indicator model' without any explanation of its general hypothesis is not helpful to the reader, especially as you do not really discuss this model later in the paper or use the label to distinguish it from other models. Either briefly explain it (preferably), or remove it.

L92 Is there a word missing after 'individual'?

L265 To me it makes more sense to integrate this information on publication bias into the reported results for the overall model.

Response to Reviewer's Comments

Overview

Overall, the reviewers found the premise of our manuscript to be an important topic and of broad interest and that the analysis to be solid and competent. A major comment from Reviewer #1 was a suggestion to include studies that investigated bare-part coloration in our meta-analysis. We respectfully disagree with inclusion of bare-part studies. In our response to Reviewer #1 below, we detail key differences between feather coloration and bare part coloration that may affect how color in these structures signal condition and that was the basis for our a-priori decision to focus on feather coloration. Reviewer #2 had major comments about the structure of the introduction and discussion and suggested helpful re-organization comments that we have incorporated in to our revised manuscript. Additionally, they asked about the effect of sex on the effect size estimates. We conducted this analysis and found that sex did not influence the magnitude of the meta-analytic means that we report (details below).

Our responses are highlighted with blue text.

Reviewer #1 (Remarks to the Author):

Review of NCOMMS-17-21773, Carotenoid metabolism strengthens the link between feather coloration and quality: a meta-analysis

Overview: In this paper, the authors assess whether metrics related to quality are more closely associated with dietary, metabolically unconverted carotenoids, or metabolically converted carotenoids in the feathers of songbird species. By performing a meta-analysis, they find support for the hypothesis that there are more robust relationships between metabolically converted carotenoids and quality than unchanged, diet-derived carotenoids. This work is interesting, and the authors are absolutely correct that the often-discussed but still contentious issue of HOW carotenoids relate to quality is an important question. I do have some questions/concerns/suggestions that I outline below.

Line 70: Relatively irrelevant to the authors' arguments, but what about carotenoids from maternal effects (e.g., transfer via yolk)?

This is an interesting comment, but we feel it is beyond the scope of our intent to evaluate the role of carotenoid oxidation of dietary pigments in signaling individual quality from feather coloration. Additionally, we only included measures of color from adult bird feathers to avoid potential confounding maternal effects such as this one that the reviewer notes.

Line 72: This is one of my larger concerns – I'm unclear why bare parts were dismissed for no explicit reason. In fact, inclusion of bare parts couldn't be more timely; please see Erik N. K. Iverson and Jordan Karubian (2017) The role of bare parts in avian signaling. *The Auk*: July 2017, Vol. 134, No. 3, pp. 587-611. <https://doi.org/10.1642/AUK-16-136.1>

Inclusion of carotenoid-based bare part coloration would 1) increase sample size and thus generalization of the broader question, and 2) greatly expand the phylogenetic breadth of samples. All but two of the studies in this meta-analysis were songbirds, and yet some of the seminal work on carotenoid-based coloration was with non-passerines. Either the scope of the work needs to be explicitly restated, or bare-parts should be included. As is, it's hard for me to accept a general

question regarding this study (see lines 99-101: “Specifically, we hypothesized that the strength of the relationship between coloration signal and individual quality is dependent on whether or not the color display involved metabolic conversion of dietary carotenoids.”) There’s no reason this hypothesis shouldn’t include bare parts.

The reviewer is correct that carotenoid signals from bare parts are a widespread and important component to our understanding of the evolution of carotenoids in social and sexual selection. However, the core hypothesis that we are addressing in this manuscript is whether carotenoid metabolism affects the strength of the relationship between measures of individual quality coloration of a single tissue: feathers. We chose to focus only on one signaling trait (feather coloration) and excluded bare parts for several reasons that we regrettably did not fully explain in the original version of our manuscript. We strongly believe that addressing the question of whether converted vs dietary carotenoids better signals quality is best investigated by eliminating as many other confounding biological factors involved in the production of carotenoid-color traits to isolate and focus on the phenomenon of interest: carotenoid oxidative metabolism. Specifically, the confounding factors of 1) blood flow enhancing coloration of bare parts from hemoglobin, 2) the role of esterification (the binding of fatty acids to carotenoids) of carotenoids used in bare part displays and 3) the differences in enzymes required for deposition of carotenoids to feathers and bare-parts would add unnecessary noise to our question of interest. Negro et al 2006 and Dwyer 2014 have shown flushing of bare part carotenoid traits is widespread and can directly obscure the role of carotenoids in producing the color signal ^{1,2}. De Blas et al 2013 has shown that the concentration of esterified carotenoids in red-legged partridge legs are better predictors of red coloration than free forms of the same carotenoids ³ but the degree of esterification of the carotenoids in the bills, skin, and eyes of species in most published studies is unknown. Feathers are exempt from both of these processes; they are not vascularized and only free form carotenoids have been detected in feathers.

Also, in the recent papers identifying CYP2J19 as the gene that codes for the ketolase enzyme in birds, it was shown that CYP2J19 alone enables red bill coloration in Zebra Finches ⁴ but CYP2J19 plus a keratin gene is needed for red feather coloration ⁵. Minimally, this shows that the integration of converted carotenoids into growing feathers involves a novel developmental pathway that is not required for bill coloration. A recent paper by Badyaev showing connections between converted and dietary carotenoids and feather shape, cited in the main document, further underscores potential key differences in the deposition of carotenoids in feathers and bare parts. This is a topic deserving of further study, but at this point, we feel that it is best to treat bare part coloration and feather coloration as arising from distinct developmental pathways.

For these reasons, we feel it is not appropriate to include bare part traits colored by carotenoids in the current analyses or manuscript.

We must emphasize that an interesting and open question is whether feathers or bare parts better signal quality, while controlling for carotenoid type! However, this question is beyond the scope of our manuscript and requires an entirely separate introduction, identification of relevant studies, data extraction, statistical analyses, and discussion than what is appropriate to conduct within a single manuscript. Also, what about other taxa? Fish and reptiles also show bare-part carotenoid coloration. This would make for an interesting analysis of how honesty of carotenoid coloration holds up across taxa. These questions require separate analyses and that are appropriate for separate manuscripts. We suggest this for future research in the revised manuscript.

1. Negro, J. J., Sarasola, J. H., Fariñas, F. & Zorrilla, I. Function and occurrence of facial flushing in

- birds. *Comp. Biochem. Physiol. - A Mol. Integr. Physiol.* **143**, 78–84 (2006).
2. Dwyer, J. F. Correlation of cere color with intra- and interspecific agonistic interactions of crested caracaras. *J. Raptor Res.* **48**, 240–247 (2014).
 3. García-de Blas, E., Mateo, R., Viñuela, J., Pérez-Rodríguez, L. & Alonso-Alvarez, C. Free and esterified carotenoids in ornaments of an avian species: the relationship to color expression and sources of variability. *Physiol. Biochem. Zool.* **86**, 483–98 (2013).
 4. Mundy, N. I. *et al.* Red Carotenoid Coloration in the Zebra Finch Is Controlled by a Cytochrome P450 Gene Cluster. *Curr. Biol.* **26**, 1435–1440 (2016).
 5. Lopes, R. J. *et al.* Genetic Basis for Red Coloration in Birds. *Curr. Biol.* **26**, 1427–1434 (2016).

Line 82: It was unclear whether or not the authors were restricting their study to songbirds, as opposed to all birds.

No, as we stated in our methods, we used all data available for carotenoid-based feather coloration in birds.

Figure 1. Please include information on whether marine birds may acquire astaxanthin from their diets, though.

Yes, the reviewer is correct. A very few species of birds, such as flamingoes, have red/pink feathers from directly consuming red carotenoids from their diet. We added a sentence to describe this in the figure caption.

Line 106: I found this line to be too vague. What is meant by system performance? Later, the authors also talk about cellular respiration (line 377), which is fine. But the language kept on switching. Are system performance, cellular pathways (line 318), and cellular processes (lines 288 and 322) all talking about the same thing? During these arguments, I kept on wondering if the authors were doing a good job about seeing the forest rather than the trees, or if they were engaging in hand-waving. I think tightening up the language may shift readers toward the former.

We agree that we did not make clear what is meant by system performance and have changed this wording to better describe our intentions. We mean function of cellular processes that are relevant factors in determining “quality”. This is a tedious topic that one of the authors has described in more detail here: Hill 2011 Ecology Letters, cited in manuscript. We will briefly summarize and cite this publication. For clarity and consistency with the rest of the manuscript, we have revised this to “individual quality” and cite Hill 2011.

Line 114: What about British versions of spelling colour and colouration?

Yes, we searched with and all possible alternative English spellings. We have made this clear in the manuscript.

Line 121: I think justifying why focusing ONLY on plumage-based colors is critical. Or – what would be even better – expanding to include bare part-based coloration.

We agree with the reviewer that we did not adequately justify our decision to exclude other carotenoid-pigmented tissues, such as bare parts. We explain our justification in lines 132 -139 of

the revised manuscript. We believe it is critical to only focus on a single tissue when examining the effect of converted vs dietary carotenoid because of the potentially confounding biological factors involved with bare part coloration such as blood flow, carotenoid esterification, and the different genes and developmental pathways involved in expression of converted carotenoid coloration in feathers.

Line 124: I didn't follow this. Why was measuring plasma concentrations of carotenoids a reason for exclusion? Did authors of papers used in the meta-analysis need to identify carotenoids in feathers, because I didn't see that statement. What about carotenoid IDs from other tissues (e.g., liver)? Did authors have to ID carotenoids in that specific study for it to be included, or did just one lab group have to ID the carotenoids in a species for any study on that species to be included? More clarity is required here.

We agree that we should have made this clearer in our manuscript. Measuring plasma concentrations of carotenoids was not a reason for exclusion. We excluded studies that turned up in our search results that did not report measures of feather coloration. Some studies from our searches reported only the relationship between plasma carotenoid concentration and measures of quality like body condition or immune measures of bird species that had carotenoid pigmented feathers. However, they did not measure feather coloration as a response. These were not included in our study because they did not report feather coloration measures. We are not concerned with carotenoid concentrations of any internal tissues because they are not directly signaled through phenotype. We did not require studies to identify the specific carotenoids present in their study species. We compiled those data ourselves based on other publications and reports. To summarize, these exclusion criteria make it clear that we are interested only in measures on feather coloration of adult birds. For an explanation of why this is our interest for this study, please see above.

Table 1. Oxidative physiology cannot be used interchangeably with immune function. I fully disagree with the authors' inclusion of AO capacity and OD in the Immune category. I'd also note that their main citation on this front (Simons et al. 2012) also separates these two factors (immune function and oxidative stress).

Yes, this is a fair point. We will rename the title of this category to "Immune function and oxidative physiology". The amount of studies that met our criteria were too few to have these two intimately related categories split and maintain statistical power to make meaningful inference. Including these two related categories together gives us an idea of how carotenoid coloration signals physiologically related processes that could impact fitness.

Simons et al 2012 was interested in what carotenoid coloration signals; either immune function, oxidative stress, or both. And they found that for the most part, they don't reliably signal either. Simons et al. 2012 split these categories and found no difference in effect size between measures of oxidative stress and all but one measure of immune function for 'trait redness'. This indicates that splitting these two related categories does not obscure overall patterns of the relationship between 'trait redness' and these categories.

Line 181: findings

Corrected

Table 2: The PDF version doesn't look right (I can't see all of the categories). Please check.

Yes, regrettably the PDF conversion made parts of our table unreadable. We will convert this table to a pdf ourselves and ensure it is readable before uploading to resubmission.

Line 261: Amounts?

We have added the amounts of residual heterogeneity (which are also presented in Table 2) to the results section.

Lines 286-289: It's a statement like this that screams for the inclusion of bare part coloration.

We have included a few sentences to describe how the focus of our analysis supports our hypothesis, but further study should examine how tissue type might also affect the relationship between color and quality. Please see above for our discussion for more on this topic and why it was not appropriate to include bare-part coloration in our study.

Lines 316-326: Some more precision here might help sell this idea. Does this apply to all cells? All organs? At all time points in the animal's life? I think I get the general idea, but would this apply only to the cells doing the converting of carotenoids? And what would those cells have to do with parasite resistance (but not immune function)? Statements like lines 320-323 sound like an a priori prediction for a positive relationship between metrics of immune function and coloration. At the very least, this statement wouldn't disambiguate between immune function and parasite resistance.

The reviewer brings up an interesting point that involves the relationship between specific cells that are the site of carotenoid metabolism, and the total population of cells in an animal. The specific location where carotenoid metabolism occurs within a bird's body is unclear. Some studies have suggested that it occurs at the integument, others posit that the liver is a key site for conversion. The idea that is now explained in theoretical papers regarding honest signaling is that there is a homeostatic state that includes the entire body of the animal. This is the animal's condition. The cellular pathways that create this state of high or low condition will also affect carotenoid conversion. The idea is that this will typify an entire organism. All of these assumptions are largely untested. The direction of future studies of condition-dependent signaling will invoke increasingly specific biochemical mechanisms that link condition to ornamentation. Our study shows that such a focus is justified.

Lines 341-347: Please be clear the oxidative physiology isn't a sub-category of immune function.

Yes, we have made this clearer in the revised manuscript but note that they are intimately linked (see Brambilla et al 2008⁶, Koch et al 2016⁷).

6. Brambilla, D. *et al.* The role of antioxidant supplement in immune system, neoplastic, and neurodegenerative disorders: a point of view for an assessment of the risk/benefit profile. *Nutr. J.* **7**, 29 (2008).

7. Koch, R. E., Josefsen, C. C. & Hill, G. E. Mitochondrial function, ornamentation, and immunocompetence. *Biol. Rev.* (2016). doi:10.1111/brv.12291

Lines 388-393: Based on which results? I'm not sure why metrics of immune function are not related to quality, but parasite resistance is.

The results of our meta-analytic model that shows converted carotenoids are better predictors of measures of quality, overall (Fig 2). The resource trade-off hypothesis predicts that whether or not carotenoids are modified should have no effect on the strength of the relationship between color and quality because it is access to carotenoids that is driving the whole relationship. The shared pathway hypothesis predicts that carotenoid metabolism can signal quality when the pathways involved in carotenoid metabolism are linked to the pathways that underlie individual condition. We have discussed this in the manuscript.

Reviewer #2 (Remarks to the Author):

The authors present a meta analysis of studies investigating the association between the richness of carotenoid colouration of bird plumage, and measures of individual quality. Overall, they find a moderate-sized, positive association that does not statistically differ from zero. However, when the data are partitioned by carotenoid type (whether dietary carotenoids are directly deposited in plumage or first metabolised and converted to ketocarotenoids) and by the measure of individual quality used, there is a strong positive association between converted (but not dietary) carotenoid colouration and measures of parasite resistance and parental/reproductive performance. The authors argue that the metabolic conversion of carotenoids could act to maintain the honesty of colour signals, which seems plausible though it would be great to see this argument developed in even greater detail if possible. Interestingly, the analysis finds no associations between carotenoid colouration and either body condition or immune function, contra popular hypotheses for the signalling function of carotenoids.

The study is a solid and competent analysis of an interesting body of literature, and the contrast it reveals between dietary and converted carotenoids seems like a promising avenue for further research. I feel that this could also be developed further in the manuscript. What was lacking for me in the discussion was consideration of why some species convert carotenoids before deposition and others do not: what selects for ornamentation with dietary carotenoids if these are not signals of quality; and why convert carotenoids if they can be directly deposited in the plumage?

Yes, this is a very interesting question indeed! Why do some birds bioconvert carotenoids and others do not? Nearly all birds have the enzyme responsible for the oxidation reactions involved in bioconversion of carotenoids (ketolase CYP2J19, expressed and function in their retinas) but only some deposit oxidized (converted) carotenoids to their external tissues. Why? This remains an open question in evolutionary biology. This question however, is far beyond the scope, or the capacity of our data, to address in this manuscript. We added a new section to the discussion that discusses recently hypothesized mechanisms for the evolution of carotenoid metabolism for feather coloration.

More generally, I think the manuscript would benefit from a more thorough treatment of the theoretical background to the study question – at the moment the introduction seems a bit light and a lot of helpful concepts only appear in the discussion. More on the main hypotheses around carotenoids as signals would be especially valuable, given you want to highlight the importance of

biochemical and physiological mechanism to these hypothesised selective processes.

We agree that we have not included some important background information in the introduction to make it easier to follow by a wide audience. We have rewritten the introduction to include key points about the role of carotenoids in sexual signaling and physiology (Lines 66-85 of the revised manuscript) following the suggestions of the reviewer. We also note that the location of Box 1 which covers additional detailed information about carotenoids is located at the very end of the manuscript file provided for review. When the article is published this may be positioned closer to the introduction to help with providing relevant background and sufficient details to the reader when reading the introduction.

The intro itself could be better organised overall. I would suggest starting with carotenoids (e.g. something like the current lines 65-70) before the current first paragraph, and moving the section that starts on line 86 up to merge with the current first paragraph. This might help to clarify your statements on the state of the literature – currently, the first para implies that there is general support for the hypothesis that carotenoid colouration signals individual quality, but the mechanisms for this are disputed, while line 92 suggests that the relationship between carotenoid expression and quality is not generally supported. You could then merge the section line 72-84 with line 94-107. Finally, the specific predictions on line 101-107 are not immediately intuitive to me and would benefit from additional explanation.

Yes, the reviewer's suggestions will make the introduction more informative and clear. We have incorporated these changes and rewritten the introduction.

Although the intro and methods do not mention sex differences in colour at all, you start the discussion with reference to the idea that carotenoid colouration in males is a signal used by females to assess mate quality. Apart from this being rather late in the manuscript to introduce new concepts, I am curious whether the studies included in your dataset used species that display sex-specific colouration, or even focused on males. Did the majority of studies measure carotenoid colour and quality in males only? If not, then sex differences seem like a possible source of heterogeneity – could you account for this in your analysis? Looking at Table 1, there are a couple of 'quality' proxies that seem quite specific to female fitness (e.g. in the Reproduction category, clutch size or lay date) which suggests to me that at least some studies measured females, and I wonder if you might find differences in the type of carotenoid (dietary or converted) in species where males only, or both sexes, express the colour ornamentation.

We have made the beginning of the discussion more broad by removing the qualifier of male. It is likely that both males and females assess coloration as a criterion in mate choice. Both males and females were assessed in these analyses. 29 effect sizes did not indicate sex (5 converted, 24 dietary), 35 effect sizes were from females (15 converted, 20 dietary), and 133 effect sizes were from males (75 converted, 58 dietary). Per the reviewer's suggestion. We analyzed how sex affects the correlation between feather color and quality overall for both converted and dietary coloration. That is, the interaction between sex and carotenoid type. We found no indication that the sex from which the effect sizes were estimated influences the magnitude of the meta-analytic mean. differences between sexes of the same carotenoid type. Dietary M vs F $\beta = 0.19$, 95% CI: -0.080 to 0.487. Converted F vs M $\beta = -0.10$, 95% CI: -0.283 to 0.07.

Altogether, I think this is an interesting study but it would benefit from some thoughtful editing to better set up the background to the study, and provide more solid logical and indirect support for your interpretation of the influence of carotenoid type.

We thank the reviewer for their helpful comments that will improve the understanding of the introduction and discussion. We have revised the introduction and discussion as suggested.

Minor:

L58 Referring to ‘the indicator model’ without any explanation of its general hypothesis is not helpful to the reader, especially as you do not really discuss this model later in the paper or use the label to distinguish it from other models. Either briefly explain it (preferably), or remove it.

We have rewritten the introduction and more broadly describe the hypothesis that carotenoid coloration is an honest signal of individual quality.

L92 Is there a word missing after ‘individual’?

Yes, we revised this sentence to read “individual quality”.

L265 To me it makes more sense to integrate this information on publication bias into the reported results for the overall model.

Yes, we agree and have moved the results of the publication bias test to the first paragraph of the results section describing the overall model.

Reviewers' comments:

Reviewer #1 (Remarks to the Author):

Review of NCOMMS-17-21773, Carotenoid metabolism strengthens the link between feather coloration and quality: a meta-analysis

Overview: In this revision, the authors have addressed the majority of my comments. I have a few thoughts on some of their revisions below.

Response to line 72:

Fair enough, and the inclusion of the text in Revision Lines 132-139 addresses these concerns.

Response to Line 82 comment:

I'm not sure how to handle this. It's true that the authors used data from all possible species, including non-passerines. However . . . there only 2 of the 21 species are non-passerines, constituting only 6 of the 197 effect sizes. The passerine-tilted dataset is apparent even from the authors' viewpoints; in the legend to figure 1, passerines are called out explicitly. So I'm not sure what the most advisable course of action is. Keeping everything as is seems a little odd, based on how phylogenetically removed penguins are. But if the data exist – why not use them? Just something to consider.

Response to Line 124 comment:

Please clarify which metrics were used. Brightness is not used, but how were Munsell and PCA data used (if they even were)?

Also, in rereading this section, I think I identified my source of confusion; there's not a clear explanation regarding how effect sizes were categorized as dietary or converted. As figure 1 shows, color alone isn't sufficient, as there are yellow converted pigments and yellow dietary pigments. So – how was it determined whether an effect size belonged to converted or dietary? This is why I originally asked questions such as "Did authors of papers used in the meta-analysis need to identify carotenoids in feathers?" This is a huge deal – how did the authors of this paper determine whether the studies they used were based upon converted or dietary pigments?

Response to table 1 comment:

I didn't find this to be a compelling counter-argument. Immune function is much more closely aligned with parasite load and resistance than it is to oxidative physiology, which can be affected by everything from circulating hormone levels to digestive processes. So combining immune function with oxidative physiology seems odd. If the authors are concerned about signaling "physiologically related processes" then they can combine immune function with parasite information. Or treat all three categories separately. Yes, oxidative physiology can be affected by the immune system. However, it can also be affected by the digestive system, or the endocrine system, or a variety of other factors. Lumping these two categories together is not needed, and a little misleading. Oxidative physiology is not a subset of the immune system. For example, glucose homeostasis is

linked to the immune system, but it would be pretty odd to lump the immune system and nutrient physiology into a single analysis . . .

Reviewer #2 (Remarks to the Author):

In their revision, the authors have greatly improved the introduction which now gives a well-justified rationale for their focal analyses and clearly lays out the main alternative hypotheses under consideration.

I appreciate the addition of the analysis investigating potential sex differences. My final comment along these lines is that it seems a little odd that the discussion paragraph lines 349-360 remains so focused on male ornamentation and female choice given that:

- it is likely that both sexes assess carotenoids in their mates (as stated by the authors in their response to my previous comment);
- some of the included studies measured carotenoids in females (which the new analysis shows do not differ from carotenoids in males in their relationship with measures of quality);

and now, having made the beginning of the discussion less specific to male ornamentation, this paragraph is the only sex-specific one in the whole paper. I do not think it would be too hard to make these statements about choice and ornamentation more general (i.e. "...are a composite of the ornamented bird and its mate. Mate choice for colouration should be stronger..."), or alternatively you could at least add the disclaimer that you refer here to male ornaments and female choice for convenience, because it is the more common pattern in birds, and because the majority of effects included in your analyses were measured in males.

Response to reviewers:

We have added a description of how we categorized carotenoid types and provided citations for each classification (lines 352 -360). A table with each species, its classification, and citations were added as Supplementary Table 4. We also performed the analyses that split oxidative measures from immune measures and also lump parasite measures into the immune category. Results from each of these iterations are provided in the supplementary information.

Reviewers' comments:

Reviewer #1 (Remarks to the Author):

Review of NCOMMS-17-21773, Carotenoid metabolism strengthens the link between feather coloration and quality: a meta-analysis

Overview: In this revision, the authors have addressed the majority of my comments. I have a few thoughts on some of their revisions below.

Response to line 72:

Fair enough, and the inclusion of the text in Revision Lines 132-139 addresses these concerns.

Response to Line 82 comment:

I'm not sure how to handle this. It's true that the authors used data from all possible species, including non-passerines. However . . . there only 2 of the 21 species are non-passerines, constituting only 6 of the 197 effect sizes. The passerine-tilted dataset is apparent even from the authors' viewpoints; in the legend to figure 1, passerines are called out explicitly. So I'm not sure what the most advisable course of action is. Keeping everything as is seems a little odd, based on how phylogenetically removed penguins are. But if the data exist – why not use them? Just something to consider.

We agree with the reviewer's concern and have conducted all analyses with only passerine species. The results are qualitatively similar. The unequal distribution of effect sizes among passerines and non-passerines could be viewed as problematic; few passerine's display carotenoid-pigmented plumage. To avoid this, we report results from only the passerine analyses. The results section, tables, and figures have been updated with the passerine-only data.

Response to Line 124 comment:

Please clarify which metrics were used. Brightness is not used, but how were Munsel and PCA data used (if they even were)?

We have clarified how color metrics were used in the Color Metrics sub section of the Methods in the revised manuscript (lines 368-370).

Also, in rereading this section, I think I identified my source of confusion; there's not a clear explanation regarding how effect sizes were categorized as dietary or converted. As figure 1 shows, color alone isn't sufficient, as there are yellow converted pigments and yellow dietary pigments. So – how was it determined whether an effect size belonged to converted or dietary? This is why I originally asked

questions such as “Did authors of papers used in the meta-analysis need to identify carotenoids in feathers?” This is a huge deal – how did the authors of this paper determine whether the studies they used were based upon converted or dietary pigments?

Yes, we agree that this is important information that we have now made clear in the revised manuscript. Additionally, we provide references for our justification to classify a birds feather coloration to be from dietary or converted carotenoids (Supplementary Table 4).

Response to table 1 comment:

I didn't find this to be a compelling counter-argument. Immune function is much more closely aligned with parasite load and resistance than it is to oxidative physiology, which can be affected by everything from circulating hormone levels to digestive processes. So combining immune function with oxidative physiology seems odd. If the authors are concerned about signaling “physiologically related processes” then they can combine immune function with parasite information. Or treat all three categories separately.

Yes, oxidative physiology can be affected by the immune system. However, it can also be affected by the digestive system, or the endocrine system, or a variety of other factors. Lumping these two categories together is not needed, and a little misleading. Oxidative physiology is not a subset of the immune system. For example, glucose homeostasis is linked to the immune system, but it would be pretty odd to lump the immune system and nutrient physiology into a single analysis . . .

We repeated our analyses with: (1) measures of oxidative physiology as its own category, and (2) measures of oxidative physiology as its own category and measures of parasite resistance lumped into the immune function category. We note this in the results and report the outcomes of these analyses in the supplemental information (Supplementary Table 2 and Table 3). Note that only 1 effect size in the oxidative physiology category is from converted carotenoids.

Reviewer #2 (Remarks to the Author):

In their revision, the authors have greatly improved the introduction which now gives a well-justified rationale for their focal analyses and clearly lays out the main alternative hypotheses under consideration.

I appreciate the addition of the analysis investigating potential sex differences. My final comment along these lines is that it seems a little odd that the discussion paragraph lines 349-360 remains so focused on male ornamentation and female choice given that:

- it is likely that both sexes assess carotenoids in their mates (as stated by the authors in their response to my previous comment);
- some of the included studies measured carotenoids in females (which the new analysis shows do not differ from carotenoids in males in their relationship with measures of quality);

and now, having made the beginning of the discussion less specific to male ornamentation, this paragraph is the only sex-specific one in the whole paper. I do not think it would be too hard to make these statements about choice and ornamentation more general (i.e. "...are a composite of the ornamented bird and its mate. Mate choice for colouration should be stronger..."), or alternatively you could at least add the disclaimer that you refer here to male ornaments and female choice for convenience, because it is the more common pattern in birds, and because the majority of effects included in your analyses were measured in males.

We have revised the discussion to be more sex general (e.g., mate, instead of male or female).